# Novel Insights to Assess Climate Resilience in Goats Using a Holistic Approach of Skin-Based Advanced NGS Technologies

**DOI:** 10.3390/ijms241210319

**Published:** 2023-06-19

**Authors:** Silpa Mullakkalparambil Velayudhan, Veerasamy Sejian, Chinnasamy Devaraj, Gundallahalli Bayyappa Manjunathareddy, Wilfred Ruban, Vinod Kadam, Sven König, Raghavendra Bhatta

**Affiliations:** 1Institute of Animal Breeding and Genetics, Justus Liebig University Giessen, Ludwigstr. 21b, 35390 Giessen, Germany; mv.silpa@gmail.com (S.M.V.); sven.koenig@agrar.uni-giessen.de (S.K.); 2Rajiv Gandhi Institute of Veterinary Education and Research, Kurumbapet, Pondicherry 605008, India; 3Centre for Climate Resilient Animal Adaptation Studies, ICAR-National Institute of Animal Nutrition and Physiology, Adugodi, Bangalore 560030, India; drcdeva@gmail.com (C.D.); ragha0209@yahoo.com (R.B.); 4ICAR-National Institute of Veterinary Epidemiology and Disease Informatics, Bangalore 560064, India; gbmpatho@gmail.com; 5Department of Livestock Product Technology, Hebbal Veterinary College, Karnataka Veterinary Animal and Fishery Sciences University, Hebbal, Bangalore 560024, India; rubanlpt@gmail.com; 6Textile Manufacturing and Textile Chemistry Division, Central Sheep and Wool Research Institute, Avikanagar, Malpura 304501, India; vinod.vjti@gmail.com

**Keywords:** DNA methylation, goats, heat stress, metagenomics, skin, transcriptomics

## Abstract

A novel study was conducted to elucidate heat-stress responses on a number of hair- and skin-based traits in two indigenous goat breeds using a holistic approach that considered a number of phenotypic and genomic variables. The two goat breeds, Kanni Aadu and Kodi Aadu, were subjected to a simulated heat-stress study using the climate chambers. Four groups consisting of six goats each (KAC, Kaani Aadu control; KAH, Kanni Aadu heat stress; KOC, Kodi Aadu control; and KOH, Kodi Aadu heat stress) were considered for the study. The impact of heat stress on caprine skin tissue along with a comparative assessment of the thermal resilience of the two goat breeds was assessed. The variables considered were hair characteristics, hair cortisol, hair follicle quantitative PCR (qPCR), sweating (sweating rate and active sweat gland measurement), skin histometry, skin-surface infrared thermography (IRT), skin 16S rRNA V3-V4 metagenomics, skin transcriptomics, and skin bisulfite sequencing. Heat stress significantly influenced the hair fiber characteristics (fiber length) and hair follicle qPCR profile (*Heat-shock protein 70 (HSP70)*, *HSP90*, and *HSP110*). Significantly higher sweating rate, activated sweat gland number, skin epithelium, and sweat gland number (histometry) were observed in heat stressed goats. The skin microbiota was also observed to be significantly altered due to heat stress, with a relatively higher alteration being noticed in Kanni Aadu goats than in Kodi Aadi goats. Furthermore, the transcriptomics and epigenetics analysis also pointed towards the significant impact of heat stress at the cellular and molecular levels in caprine skin tissue. The higher proportion of differentially expressed genes (DEGs) along with higher differentially methylated regions (DMRs) in Kanni Aadu goats due to heat stress when compared to Kodi Aadu goats pointed towards the better resilience of the latter breed. A number of established skin, adaptation, and immune-response genes were also observed to be significantly expressed/methylated. Additionally, the influence of heat stress at the genomic level was also predicted to result in significant functional alterations. This novel study thereby highlights the impact of heat stress on the caprine skin tissue and also the difference in thermal resilience exhibited by the two indigenous goat breeds, with Kodi Aadu goats being more resilient.

## 1. Introduction

Climate change is an inevitable fact that is causing worry across all sectors on the global front. Among these sectors, the livestock industry is of high relevance for its significant contribution towards the global economy [1]. Alterations in climatic conditions lead to a number of environmental stresses in livestock, among which heat stress is of major concern. Though animals possess a number of adaptive mechanisms to combat the challenging environmental conditions, the challenges that exceed their thermoregulatory mechanisms can be stated to be heat stress [2]. The impact of heat stress in livestock is devastating in that it not only results in severe production and economic losses but is also a concern from the perspective of animal welfare. 

The response to heat stress varies among species, breeds, and individuals. Recent literature reviews and modeling approaches clearly projected goats to be the ideal climate animal model [3,4,5]. Their ability to survive and produce optimally in extremely harsh environments in addition to possessing some vital climate-resilient traits such as excellent thermo-tolerance, disease resistance, and ability to survive on limited pastures make them the go-to species [5]. Advances in the field of biotechnology have facilitated several researchers across the globe to obtain a quicker and relatively more accurate way to identify the impact of heat stress in animals, assess the animal’s adaptability to heat stress, and also identify novel biomarkers that could be used for future breeding strategies. Studies on the response of goats to heat stress [6] and ones assessing their adaptation to heat stress [7], impact on meat production [8,9], milk production [10], and also immune status [10,11] have been fairly reported. However, one aspect that has been the least explored in any of the livestock species is skin and its associated changes pertaining to goat adaptation. Skin, apart from being the largest and most widely distributed organ, is the first organ to be exposed to environmental alteration. Skin acts as a protective barrier that also has a crucial role in shielding the animal from heat stress. Hence, it is imperative to study this organ in detail and understand the alterations occurring within it during heat-stress exposure to understand the hidden intricacies of adaptive mechanisms that could pave the way for identifying climate-resilient breeds. 

With this background, a study was designed to understand the implications of heat stress on the skin-related adaptive responses in two indigenous goat breeds, i.e., Kanni Aadu and Kodi Aadu. This novel study is the first of its kind in attempting to elucidate the heat-stress responses of a number of hair- and skin-based traits along with the applications of advanced next-generation sequencing technologies such as skin metagenomics, skin transcriptomics, and skin epigenetics (bisulfite sequencing). Additionally, this study also compares the adaptive capabilities of two indigenous goat breeds using such a holistic skin-based approach, which is also the first to be reported ever in any livestock species.

## 2. Results

### 2.1. Simulation of Comfort and Heat-Stress Environment within the Climate Chambers

The average temperature humidity index (THI) values obtained in the morning were comparable for both the control/thermo-neutral zone (TNZ) (71.34 ± 0.09) and heating (72.13 ± 0.07) chambers. A highly significant (*p* < 0.01) difference in the afternoon THI values were observed for the TNZ and heating chambers, which observed average values of 69.42 ± 0.05 and 94.76 ± 0.45, respectively.

### 2.2. Hair Fiber Analysis

On assessing the hair characteristics, a significant difference in the fiber diameter (*p* = 0.00) and staple length (*p* = 0.05) was observed between the four groups, namely KAC, KAH, KOC, and KOH (Table 1).

Though heat stress did not have an immediate impact on fiber diameter, breed-specific differences were observed wherein the Kodi Aadu goats had thicker hair (82.93 ± 3.18 µm in KOC and 84.24 ± 3.97 µm in KOH) when compared to Kanni Aadu goats (68.00 ± 2.43 µm in KAC and 64.33 ± 1.49 µm in KAH). The hair length was comparable between the KAC (3.77 ± 0.03 cm) and KAH (3.53 ± 0.12 cm), while a significant difference in the length was observed between KOC (3.72 ± 0.27 cm) and KOH (3.11 ± 0.17 cm), the later having shorter staple length (Table 1). Additionally, Table 2 depicts the correlation coefficient between breed, group, and THI with hair characteristics, hair cortisol, hair follicle qPCR, sweating, skin histometry, and skin-surface IRT. Correlation analysis revealed a significant (*p* < 0.01) positive correlation between fiber diameter and breed, group, hair follicle HSP70 and HSP110 mRNA expression, and skin epithelial height. Likewise, staple length was found to be negatively correlated with group (*p* < 0.05), THI (*p* < 0.05), hair follicle HSP70 mRNA expression (*p* < 0.05), HSP90 (*p* < 0.01), sweating rate (*p* < 0.05), active sweat gland number (*p* < 0.01), skin epithelial height (*p* < 0.05), and skin-surface IRT (*p* < 0.05).

### 2.3. Hair Cortisol Estimation

The impact of heat stress on hair cortisol levels is depicted in Table 1. The hair cortisol levels in the control and heat-stress groups of Kanni Aadu and Kodi Aadu goats were found to be comparable (*p* > 0.05). Further, the hair cortisol levels were found to have no significant correlation with any of the estimated variables (Table 2).

### 2.4. Hair Follicle qPCR Analysis

The differences in the hair follicular mRNA expression profiles of *HSP70*, *HSP90,* and *HSP110* are depicted in Table 1. The hair follicular *HSP70* mRNA relative expression was comparable (*p* > 0.05) in KAC (1.00 ± 0.08 fold change) and KAH (0.99 ± 0.12 fold change), while that of Kodi Aadu goats was significantly (*p* < 0.05) down-regulated in KOH (0.52 ± 0.19 fold change) when compared to KOC (1.00 ± 0.13 fold change). However, the hair follicular relative expression profile of *HSP90* mRNA was significantly altered due to heat stress both in Kanni Aadu (*p* = 0.01) and Kodi Aadu (*p* = 0.00) goat breeds. The relative expression of *HSP90* mRNA in the hair follicle of KAC, KAH, KOC, and KOH groups was 1.00 ± 0.11; 0.63 ± 0.07; 1.00 ± 0.01; and 0.26 ± 0.10, respectively. Similar to HSP70, heat stress did not have a significant (*p* > 0.05) effect on the expression pattern of hair follicular *HSP110* mRNA in KAC and KAH, while the same gene was significantly down-regulated in KOH (0.26 ± 0.10 fold change) when compared to KOC (1.00 ± 0.01 fold change). The correlation analysis revealed the hair follicle mRNA expression of all the three selected HSPs was significantly correlated with breed, group, active sweat gland number, skin epithelial height, and total sweat gland number (Table 2). Among the three HSPs, hair follicular expression of *HSP90* was found to be significantly correlated with THI (*p* < 0.01), with a positive correlation coefficient of 0.60, while that of *HSP70* and *HSP110* was not significant. Likewise, the mRNA expression of HSP90 was significantly correlated with the sweating rate (*p* < 0.01) and IRT recorded at all five regions (*p* < 0.01) of the caprine species.

### 2.5. Sweating Rate and Active Sweat Gland Estimation

The animals housed in the TNZ chamber (KAC and KOC) showed no sweating, while animals subjected to heat stress exhibited a significantly higher sweating rate when compared to their respective controls; KAH: 2.80 ± 0.37 gm^−2^ h^−1^ and KOH: 2.82 ± 0.37 gm^−2^ h^−1^, respectively. The sweating rate was observed to be significantly correlated with group (*p* < 0.05) and also had a high positive correlation with THI (*p* < 0.01; correlation coefficient: 0.99; Table 2). Apart from the correlation with the previously mentioned variables, sweating in the caprine breeds showed a strong positive correlation with the skin-surface temperatures recorded using IRT for all five regions (*p* < 0.01; correlation factor: 0.98).

For the first time ever, the number of active sweat glands in goats was estimated (Table 1). Heat stress had a highly significant influence on the number of active sweat glands in goats (*p* = 0.00). No sweat glands were activated in the control group animals (KAC: 0.00 ± 0.00 glands/cm^2^ and KOC: 0.00 ± 0.00 glands/cm^2^), while heat stress significantly activated the sweat glands in both the heat-stress groups; KAH: 0.03 ± 0.01 glands/cm^2^ and KOH: 0.06 ± 0.01 glands/cm^2^. Similar to sweating rate, active sweat gland number in goats was significantly correlated with group (*p* < 0.01), THI (*p* < 0.01; 0.82 correlation coefficient), and skin-surface IRT across five regions (*p* < 0.01; correlation coefficient of 0.82 and above) (Table 2).

### 2.6. Skin Histology

To further assess the impact of heat stress on caprine skin, the skin histology was analyzed to assess the skin epithelial height and number of sweat glands (Table 1). Heat stress was observed to have a significant influence on the skin epithelial height (*p* = 0.00). The skin epithelium of animals in heat-stress groups was significantly higher than that of their respective controls. Additionally, KOH group animals were found to have the highest skin epithelium when compared to all other groups. The skin epithelial height of KAC, KAH, KOC, and KOH groups was 15.62 ± 0.23 µm, 16.84 ± 0.18 µm, 18.15 ± 0.49 µm, and 22.94 ± 0.60 µm, respectively. The number of sweat glands between the four experimental groups was observed to be highly significant (*p* = 0.00; Table 1). The number of sweat glands in KAC, KAH, KOC, and KOH groups was 421.67 ± 3.53 glands/cm^2^, 422.75 ± 1.83 glands/cm^2^, 429.33 ± 2.14 glands/cm^2^, and 431.42 ± 1.97 glands/cm^2^, respectively. The number of sweat glands between the control and heat-stress group of each breed was comparable. The correlation analysis between the skin histological variables and most of the other recorded variables is depicted in Table 2. Skin epithelial height and sweat gland number were significantly correlated with breed (*p* < 0.01) and group (*p* < 0.01). Only skin epithelial height was observed to have a significant correlation with THI (*p* < 0.05) and skin-surface IRT (*p* < 0.05).

### 2.7. Infrared Thermography of Caprine Skin

The skin-surface temperature was recorded using the infra-red thermal imager on five regions on the goats: the eye, forehead, flank, and back and front legs (Table 1; Appendix A). Heat stress had a highly significant (*p* = 0.00) influence on the skin-surface temperature in both the goat breeds. Higher surface temperature was recorded on the eye, flank, forehead, front leg, and lastly the back region in all the four experimental groups. The eye temperatures of 36.93 ± 0.11 °C, 40.96 ± 0.12 °C, 36.88 ± 0.15 °C, and 41.14 ± 0.09 °C were recorded in KAC, KAH, KOC, and KOH groups, respectively, and for the flank were 30.68 ± 0.26 °C, 40.78 ± 0.22 °C, 31.34 ± 0.27 °C, and 40.43 ± 0.13 °C, respectively. The IRT recordings across all the five animal body regions revealed significantly higher surface temperatures for goats housed in the heating chambers when compared to those in TNZ. The results of the correlation analysis between the IRT readings with other variables are depicted in Table 2. Skin-surface temperature was observed to be significantly correlated with group (*p* < 0.05), THI (*p* < 0.01, correlation coefficient: 0.99), and the previously stated traits.

### 2.8. Skin 16S rRNA V3-V4 Metagenomics

Sequencing the 16s V3-V4 rRNA gene of the caprine skin’s microbiota generated a total of 4,689,622 high-quality sequences with an average of 195,400.92 reads per sample. The diversity of the skin microbiome assessed based on its richness and evenness was estimated using alpha diversity indices such as Chao1, ACE, Shannon, Simpson, InvSimpson, and Fisher indices (Appendix A). All the estimated alpha diversity indices except for Fisher were found to have higher values for heat-stress groups (KAH and KOH) when compared to their respective control groups (KAC and KOC). The Fisher’s index depicted an increased value for KAH when compared to KAC, while that of KOH was lower than KOC. 

For the first time ever, the impact of heat stress on skin microbial diversity was assessed in goats. Taxonomic analysis revealed the presence of nearly 37 microbial phyla on the caprine skin, with the most abundant phyla belonging to *Bacteroidetes*, *Firmicutes*, *Proteobacteria*, *Actinobacteria, Spirochaetes*, *Fibrobacteres*, *Verrucomicrobia*, *TM7*, *Lentisphaerae,* and *Tenericutes*. At the phylum level, the skin microbiome of both the breeds, namely Kanni Aadu and Kodi Aadu, was found to be altered due to heat stress (Figure 1).

It was interesting to note that a majority of the altered top 10 microbial phyla had reduced relative abundance in KOH when compared to KOC, while contrast was observed in Kanni Aadu goats (Table 1). In comparison to KOC, the goats in the KOH group were found to have lower relative abundance of the skin microbial phyla *Bacteroidetes* (37.10% in KOC to 31.38% in KOH), *Proteobacteria* (20.21% to 17.12%), *Spirochaetes* (2.52% to 1.58%), *Verrucomicrobia* (2.30% to 1.49%), and *TM7* (1.04% to 0.69%). Among these phyla, only *Proteobacteria* was observed to have an evident reduction in its relative abundance in KAH (18.54%) when compared to KAC (31.86%), while the rest of the above-mentioned phyla either had increased abundance or remained nearly unaffected. Further, in Kodi Aadu goats, heat stress was observed to cause an evident increase in the relative abundance of *Actinobacteria* (6.02% to 17.48%). The remaining phyla, i.e., *Firmicutes*, *Fibrobacteres*, *Lentisphaerae,* and *Tenericutes*, exhibited negligible alteration due to heat stress in Kodi Aadu goats. In Kanni Aadu goats, heat stress resulted in an increased abundance of the phyla *Bacteroidetes* (31.86% in KAC to 33.23% in KAH), *Firmicutes* (21.40% to 29.00%), *Actinobacteria* (7.67% to 12.44%), and *TM7* (0.61% to 1.23%) when compared to their control. Additionally, the relative abundance of the phyla *Proteobacteria* and *Fibrobacteres* was found to have reduced drastically in KAH (18.54% and 1.04%, respectively) when compared to KAC (31.86% and 2.40%, respectively). Apart from all this, the relative abundance of the phyla *Spirochaetes*, *Verrucomicrobia*, *Lentisphaerae,* and *Tenericutes* was least-affected due to heat stress in Kanni Aadu goats. 

Nearly 347 microbial genera were identified from the caprine skin tissues, among which *Prevotella*, *Acinetobacter*, *Psychrobacter*, *Ruminococcus*, *Corynebacterium*, *Macrococcus*, *Kocuria*, *Fibrobacter*, *Treponema,* and *Staphylococcus* represent the top 10 most abundant caprine skin microbial genera. Heat stress was found to cause a notable alteration in the skin microbiota in goats at the genus level (Figure 1). Further, the response exhibited due to heat stress varied between Kanni and Kodi Aadu breeds (Figure 1). In Kanni Aadu goats, heat stress drastically reduced the relative abundance of *Acinetobacter* (33.25% in KAC to 17.27% in KAH), *Psychrobacter* (17.35% to 9.54%), *Macrococcus* (6.94% to 3.49%), and *Fibrobacter* (4.40% to 2.26%). In addition to this, heat stress also increased the relative abundance of *Prevotella*, *Ruminococcus, Kocuria,* and *Staphylococcus* in KAH (32.99%, 9.84%, 7.96%, and 7.42%, respectively) when compared to KAC (24.98%, 2.30%, 1.76%, and 0.30%, respectively). In Kodi Aadu goats, the relative abundance of *Psychrobacter*, *Corynebacterium*, *Kocuria,* and *Staphylococcus* was increased as a consequence of heat stress from 9.97% in KOC to 15.92% in KOH, 3.28% to 13.59%, 2.26% to 4.51%, and 0.61% to 2.18%, respectively. The genera *Acinetobacter* was found to have a drastic reduction in the KOH group (8.02%) when compared to its control group, KOC (26.02%). Likewise, heat stress also reduced the relative abundance of *Treponema* and *Ruminococcus* from 4.20% and 10.58%, respectively, in KOC and to 2.07% and 9.96%, respectively, in KOH. 

Through taxonomic analysis, nearly 221 microbial species were identified in the caprine skin tissue, among which the top 10 most abundant species were *Acinetobacter lwoffii*, *Prevotella ruminicola*, *Psychrobacter sanguinis*, *Macrococcus caseolyticus*, *Fibrobacter succinogenes*, *Psychrobacter pulmonis*, *Kocuria rhizophila*, *Ruminococcus flavefaciens*, *Ruminococcus albus,* and *Micrococcus luteus*. At the species level also, a notable alteration in the relative abundance was observed in caprine skin as a consequence of heat stress (Figure 1). 

Additionally, the Venn diagram depicting the unique and common taxa among the four groups, i.e., KAC, KAH, KOC, and KOH, is depicted in Appendix A. Heat stress, therefore, was observed to have a significant impact on the skin microbial profile in goats. Being a first time report, an additional analysis on the relative abundance among the taxonomically classified microbes was performed to determine the singular effect of heat stress on the skin microbiome of Kanni Aadi and Kodi Aadu goats by LEfSe. On comparing the KAC and KAH group (Figure 2), a total of 16 skin microbial genera were observed to significantly distinguish the two groups, while for the Kodi Aadu goats, 21 genera were identified as the distinguishing features between control and heat-stress conditions (Figure 2).

PICRUSt analysis predicted the functional pathways that were associated with the obtained skin microbiome (Figure 3). 

The KEGG pathway “Metabolism”, which was among the most abundant functional pathway, had relatively lower abundance in both the heat-stress groups (KAH and KOH) in comparison to their respective control groups (KAC and KOC). Additionally, a breed variation may also be highlighted wherein Kodi Aadu goats in general had higher abundance of the functional pathways (KEGG and COG) when compared to Kanni Aadu goats; this could further substantiate the genetic differences between the two breeds.

### 2.9. Skin Transcriptomics Analysis

The read alignment statistics for all the 12 skin-tissue samples are depicted in Appendix A. The DEG analysis revealed a total of 7993 and 2036 genes significantly expressed (log2 fold change ≥ 1 and *p*-value ≤ 0.05) in Kanni Aadu and Kodi Aadu goats due to heat stress (KAC_vs_KAH and KOC_vs_KOH, respectively; Table 3). Among these, heat stress stimulated the expression of 4237 and 302 DEGs, while 3756 and 1734 DEGs were down-regulated in Kanni Aadu and Kodi Aadu goats, respectively. The volcano plots, depicting an overview of the DEGs for Kanni Aadu and Kodu Aadu breeds on exposure to heat stress (control vs. heat stress), show an evident variation between the two breeds (Appendix A). The overlaps among DEGs with significant expression in each of the group-wise comparisons (KAC vs. KAH; KOC vs. KOH; KAC vs. KOC; and KAH vs. KOH) are depicted in the UpSet plot (Figure 4).

The list of the top 20 genes that were differentially expressed in Kanni Aadu and Kodi Aadu skin (10 up-regulated and 10 down-regulated) due to heat stress is depicted in Appendix A. Apart from this, the expression of certain genes involved in classical heat stress and skin- and coat-color-associated pathways is depicted in Table 4. The majority of these genes had contrasting expression profiles in the two goat breeds during heat stress.

The gene ontology analysis revealed 28 and 24 functional annotations in Kaani Aadu and Kodi Aadu goats, which were grouped as either cellular component (CC), molecular function (MF), or biological process (BP) (Figure 5).

While a number of GO terms were common for the two breeds, there were also some GO terms uniquely altered in each breed. However, among the several annotated GO terms, NADH dehydrogenase (ubiquinone) activity (MF) and oxygen transporter activity (MF) were significantly (*p* < 0.05) enriched in Kanni Aadu goats. Meanwhile, in Kodi Aadu goats, intermediate filament (CC), keratin filament (CC), mitochondrion (CC), NADH dehydrogenase (ubiquinone) activity (MF), and structural molecule activity (MF) were significant. Appendix A depicts the a few of the significant GO terms in Kanni Aadu and Kodi Aadu goats with their DEG details. The GO analysis also depicted immune response (BP) and a few more immune-related biological responses to be significantly enriched in Kanni Aadu goats. Genes related to immune response included *IFNG*, *TNFSF13B*, *IL4*, *CSF2*, *IL1B*, *CXCL10*, *BPI*, *IL2*, *TNF*, *IL12A*, *CD4,* and *FASLG*. Among these genes, apart from *CXCL10* and *TNF*, the rest of the genes were significantly down-regulated in heat-stressed Kanni Aadu goats when compared to their control. In Kodi Aadu goats, the intermediate filament (CC) and keratin filament (CC) GO consisted of the following genes: *KRTAP3-1*, *KRTAP15-1*, *KAP8*, *KRT25*, *KRT27,* and *KRTAP11-1*. 

The KEGG analysis revealed a total of 139 and 80 pathways in Kanni Aadu and Kodi Aadu goats. Among these, several pathways were common for the two breeds, while some were unique to each breed (Table 5).

Some of the significant (*p* < 0.05) KEGG pathways common to both breeds included metabolic pathways, oxidative phosphorylation, RNA transport, protein processing in the endoplasmic reticulum, biosynthesis of amino acids, RNA polymerase, and many more. Some of the significant KEGG pathways unique to Kanni Aadu goats included neuroactive ligand–receptor interaction, ribosome, and antigen processing and presentation. Moreover, ribosome biogenesis in eukaryotes and steroid biosynthesis were a few of the unique KEGG pathways among several others in Kodi Aadu goats.

### 2.10. Skin Bisulfite Sequencing

The read alignment statistics for all 24 skin-tissue bisulfite sequencing analyses are depicted in Table 6. 

On assessing the overview of the genomic DNA methylation, 5.97%, 5.90%, 5.69%, and 5.61% of the genomic C sites were found to be methylated in KAC, KAH, KOC, and KOH groups. Furthermore, the DNA methylation in goats was observed to be broadly of three types: CG, CHG, and CHH (wherein H stands for A, C, or T). The genome-wide CG, CHG, and CHH methylation levels for the four experimental groups are depicted in Table 6. 

It can be observed that among the three methylation types, the CG methylation was most abundant across the genome in all four experimental groups. Hence, we assessed the hypermethylation at the CGI (CG islands) regions and checked its distribution across the genomic functional regions (Appendix A). It was observed that majority of the hypermethylated CGI were located at intergenic sites, followed by intron and exon for all four groups. A very small proportion of the hypermethylated CGI was distributed across the transcription termination site (TTS) and promotor regions. Though a similar trend in distribution of hypermethylated CGI was observed, a mild variation in the level of methylation at the functional region among KAC, KAH, KOC, and KOH was evident.

To assess the impact of heat stress at the epigenetic level, differential methylation analysis was performed, comparing the control and heat-stress groups of both the goat breeds to identify the differential methylated regions (DMRs) and differentially methylated genes (DMGs) and also investigate its functional analysis (Table 3). A total of 50,560 DMRs (50,545 CpG DMRs, 7 CHG DMRs, and 8 CHH DMRs) were identified in Kanni Aadu goats when subjected to heat stress (KAC_vs_KAH). In Kodi Aadu goats, 40,648 DMRs (40,469 CpG DMRs, 164 CHG DMRs, and 15 CHH DMRs) were identified due to heat stress (KOC_vs_KOH). Among these, 25,178 and 19,657 DMRs were hypermethylated in Kanni Aadu and Kodi Aadu goats, respectively. Further, 25,382 and 20,991 DMRs were hypomethylated due to heat stress in Kanni Aadu and Kodi Aadu goats. Irrespective of the methylation type, a major proportion of the DMRs were located in the intergenic regions, followed by introns, with the least distribution in the exons, promotor, and TSS (Appendix A).

On annotating the DMRs to the databases, a total of 14,646 DMGs were identified to be significantly methylated in Kanni Aadu goats, while the total for Kodi Aadu goats was 13,388 DMGs (Table 3). In Kanni Aadu goats, 7336 DMGs were hypermethylated, while 7310 DMGs were hypomethylated due to heat stress. Likewise, in Kodi Aadu goats, heat stress resulted in hypermethylation of 6507 DMGs, while 6904 DMGs were identified to be hypomethylated (Table 3).

The COG analysis of the CpG DMRs for Kanni Aadu and Kodi Aadu goats is depicted in Appendix A. Nearly similar distributions of the DMRs across the COG classes were observed for the two breeds. Additionally, most of the DMRs were observed to be under the category “Signal transduction mechanisms”; “General function prediction only”; “Posttranslational modification, protein turnover, chaperones”; “Translation, ribosomal structure and biogenesis”; and many more. 

Furthermore, GO analysis identified a total of 27 and 18 terms in Kanni Aadu and Kodi Aadu (Figure 6).

Some of the significantly enriched BP terms in Kanni Aadu goats included single-organism process (GO:0044699), single-organism cellular process (GO:0044763), biological regulation (GO:0065007), regulation of biological process (GO:0050789), regulation of cellular process (GO:0050794), and so on, while molecular transducer activity (GO:0060089), receptor activity (GO:0004872), signal transducer activity (GO:0004871), and signaling receptor activity (GO:0038023) were among a few significant MF terms in Kanni Aadu goats. 

In Kodi Aadu goats, the significant BP terms included regulation of multicellular organismal process (GO:0051239) and response to endogenous stimulus (GO:0009719). On the other hand, molecular transducer activity (GO:0060089), receptor activity (GO:0004872), signal transducer activity (GO:0004871), signaling receptor activity (GO:0038023), and transmembrane receptor activity (GO:0099600) were among the predominant MF terms (Figure 6). Further, KEGG analysis revealed 97 pathways in Kanni Aadu and 90 pathways in Kodi Aadu goats (Figure 7).

Among these, an overlap of 72 pathways was observed between the two breeds, and 25 pathways were unique to Kanni Aadu and 18 pathways for Kodi Aadu. In Kanni Aadu goats, a major proportion of the DMGs were under “Metabolic pathways” (657 DMGs). It was interesting to notice that none of the DMGs (having a KEGG pathway annotation) fell under the “Metabolic pathways” in Kodi Aadu goats. 

### 2.11. Linking Skin Transcriptomics and Epigenetics Analysis

To facilitate a better understanding of the changes occurring at the molecular level in the skin, a deeper screening of genes was performed. Of the total 50,560 DMGs and 7993 DEGs in Kanni Aadu, 4234 genes were found to be differentially methylated as well as differentially expressed (Figure 8). 

Similarly, in Kodi Aadu, 758 genes were differentially methylated as well as differentially expressed among the 40,648 DMG and 2036 DEGs. The methylation status and expression profile of these genes are depicted in the Figure 8. Upon further analyzing, of the 4234 (Kanni Aadu) and 758 (Kodi Aadu) genes (that were both differentially expressed and methylated) in the two breeds, 338 genes were identified to be common between Kanni Aadu and Kodi Aadu goats. The differential expression and methylation profile of these genes among the two breeds are depicted in Figure 8. It was interesting to observe that a major proportion of these 338 genes that were differentially methylated were up-regulated in Kanni Aadu goats, while the same were down-regulated in Kodi Aadu goats. Additionally, this list of genes that were both differentially methylated as well as differentially expressed included a number of hair follicle/hair growth/hair characteristics-related genes such as *KRT1*, *KRT10*, *KRT23*, *KRT35*, *COL1A1*, *COL1A2*, *COL2A1*, *COL3A1*, *FOS*, *FOSL2*, *KLF10*, *KLF13*, *KLF17,* and many more. Likewise, a few of the heat-shock protein genes (*HSF1*, *HSF2BP*, *HSP70.1*, *HSPA8*, *HSPB1*, *HSPB3*, and *HSPB8*) and immune-response-associated genes (*IL2*, *IL4*, *IL10*, *IL1B*, and *IL15*) were also observed. Apart from signifying the impact of heat stress at the molecular and cellular level in the skin tissue, a contrasting expression of these genes was observed between the two breeds. Most of these genes were up-regulated in heat-stressed Kanni Aadu goats, with the same being down-regulated in Kodi Aadu goats.

## 3. Discussion

For the first time ever, a comprehensive study was conducted to assess the impact of heat stress on caprine skin in addition to evaluating the difference in adaptability between the two indigenous goat breeds. Several hair- and skin-related variables were recorded while subjecting Kanni Aadu and Kodi Aadu goats to heat stress. While the results obtained revealed the significant influence of heat stress on caprine skin, an evident breed variation was also observed. 

In the current study, THI was calculated as per McDowell [12], which demarcates certain threshold limits for stress. As per this equation, a THI value below 72 is considered comfortable, while values above 78 indicate the animal to be under extreme stress. The highly significant difference in the average afternoon THI values in TNZ (69.42 ± 0.05) and the heating (94.76 ± 0.45) chambers in the current study highlights the efficiency of the controlled-climate chambers to induce heat stress in goats. Furthermore, the degree of heat stress induced due to a THI value as high as 94.76, which is way above the threshold limit of 78, must also be noticeable. Studies conducted by a number of researchers have also reported goats subjected to THI above 78, as per McDowell [12], to be under severe heat-stress conditions [7,8,9].

The anatomical characteristics of hair such as its length and diameter play a crucial role in defining an animal’s adaptive profile to heat stress [13]. In the present study, heat stress was observed to have a significant influence on fiber length (*p* = 0.05) in goats, while in the case of fiber diameter, the breed effect was more significant than the treatment. However, when looking into both these variables, it may be observed that the KOH group had shorter and thicker hair when compared to Kanni Aadu goats. This could add to the adaptive potential of Kodi Aadu goats, as previous reports have stated that animals with short, thick, and well-seated hair have an advantage to cope with warmer environments, AS these characteristics enable air circulation to cool the skin [13,14,15].

In a study led by Dulude-de Broin et al. [16], hair cortisol was represented to be a valid biomarker for the HPA axis activity in goats. In the present study, the hair cortisol levels remained unaltered due to heat stress. Apart from reflecting the thermal tolerance of Kanni Aadu and Kodi Aadu goats to the extremely severe heat stress of 94.76 THI, this also highlights their lesser dependency on the cortisol-dependent heat-stress-amelioration pathway.

HSPs are among the classical molecular chaperones that are released in cells as a response to environmental and oxidative stress [17]. The expression profile of several HSP genes have been explored in goats under exposure to heat stress [7,18,19]. However, to the best of our knowledge, this is a first-time report on hair follicular *HSP* gene expression profiling in goats. In the current study, the expression profiles of *HSP70*, *HSP90,* and *HSP110* were assessed in caprine hair follicles. Among the three genes, the relative expression of *HSP70* and *HSP110* was not significantly altered due to heat stress in Kanni Aadu goats, while the hair follicular relative expression of *HSP70* was significantly down-regulated in KAH when compared to KAC. In Kodi Aadu goats, the hair follicular expression profile of all the three genes, namely *HSP70*, *HSP90,* and *HSP110,* was significantly down-regulated in KOH when compared to KOC. Kim et al. [20] identified the hair follicular HSP gene expression profile as a novel indicator of heat stress in Hanwoo calves both under external environmental conditions and climate-controlled chambers. At this juncture, the thermo-tolerance of Kanni Aadu and Kodi Aadu breeds may be appreciated, as the *HSP* genes were either comparable or down-regulated in the heat-stress group when compared to the respective controls. Having said that, the significant down-regulation of all the three *HSPs* in Kodi Aadu goats could reflect their better resilience to heat stress than Kanni Aadu goats. Significantly lower expression of HSP genes was stated to indicate the sub-threshold level of stress induced in animals [21]. 

Though predominantly a panting type of species, goats possess considerable ability to sweat. Sweating along with the enhanced respiratory activities together facilitate effective evaporative heat-loss mechanisms in goats [22]. The significantly higher sweating rate in Kanni Aadu and Kodi Aadu goats when subjected to heat stress clearly indicates the activation of cutaneous evaporative cooling in goats. A significant difference in the sweating rate was observed in Canindé goats by Ferreira et al. [22] during the stress (from 11 a.m. to 3 p.m.) and post-stress period (from 4 p.m. to 10 a.m.). The authors credited this increased sweating rate to the activation of the latent evaporation system in goats. Likewise, Maurya et al. [23] also observed increased skin-sweating rates in Malpura sheep exposed to heat stress when compared to the control group animals. Estimation of active sweat glands could be considered another vital tool to assess heat-stress impact in livestock, with the added advantage of being a non-invasive approach. As a novel approach, the active sweat glands were estimated in goats; moreover, to the best of our knowledge, this is the first of its kind of estimation in any livestock species. Concordant with the sweating rate analysis, animals under heat stress had a significantly higher number of activated sweat glands than their respective control group.

The skin histometry of Kanni Aadu and Kodi Aadu goats revealed both breed-associated and heat-stress-induced alterations. Though no experimental results linking this trait with heat stress have been reported in goats earlier, the increased skin epithelial height in KAH and KOH groups could indicate increased surface area. This can thereby aid in increased cutaneous evaporation in the heat-stressed animals. Furthermore, the higher skin epithelial height in KOH goats could also reflect their better efficiency to adapt to the stress. The number of sweat glands estimated through skin histological sections was comparable for the control and heat-stress groups of both the breeds. However, the significantly higher number of sweat glands in KOH when compared to KAH very likely to appends the thermal-tolerance ability of Kodi Aadu goats. Additionally, it is very interesting to observe that the skin epithelium significantly correlated with most of the traits.

The skin-surface temperature that was recorded using the infrared thermography is yet another vital non-invasive tool of great importance for its advanced applications in heat-stress studies. In the present study, the skin-surface temperatures recorded in the afternoon across all the five regions were significantly higher in KAH and KOH when compared to KAC and KOC. This is in accordance with the study by Aleena et al. [7], who also recorded significantly higher skin-surface temperatures in three indigenous goat breeds upon exposure to heat stress when compared to their control groups. 

The skin also is a complex ecosystem wherein it hosts diversified microbiome. The microbial alterations in skin as a consequence of heat stress have not been explored so far in any livestock species. However, this is of equal importance, especially from the perspective of innate and adaptive cutaneous response [24]. On assessing the skin metagenomics profiles of Kanni Aadu and Kodi Aadu goats, heat stress can certainly be proven to have a significant impact on altering the skin microbiome. The relative abundance of skin microbes based on their taxonomy revealed further differences in the heat-stress response exhibited by the two breeds. The LEfSe analysis that depicted the significant microbial genera between the control and heat-stress groups portrayed an evident difference between the two goat breeds. The higher number of significantly altered skin microbial genera in the heat-stress group of Kanni Aadu goats compared to the Kodi Aadu group could reflect the better ability of the latter to withstand environmental perturbations. Furthermore, the impact of alterations of the skin microbiome due to heat stress was also reflected at the functional level based on the PICRUSt analysis. This additionally depicted a breed-associated impact wherein the Kodi Aadu in general had an increased abundance of microbes associated with the functional pathways predicted.

Though there are a few studies exploring the skin transcriptomics in livestock, this is the first study to assess the impact of heat stress on skin tissue. The transcriptomics profile revealed a drastic difference in the number of DEGs that were significantly expressed in the skin tissue due to heat stress between the two breeds. More DEGs were observed to be significantly expressed in Kanni Aadu goats (7993 DEGs) than Kodi Aadu goats (2036 DEGs). Additionally, it was also interesting to observe that in Kanni Aadu goats, a relatively higher number of DEGs were up-regulated, while contrastingly, in Kodi Aadu goats, a major proportion of the significant DEGs were down-regulated. This, in the very first instance, can reflect the better thermal-resilience potential of Kodi Aadu goats when compared to Kanni Aadu. A similar trend of more genes being significantly down-regulated in the skin of tropically adapted White Fulani (WF) cattle when compared to the temperate breed Angus (AG) was reported by Morenikeji et al. [25]. Further, the marked difference in the DEG numbers between the two breeds could also point out the coat-color-related benefit in Kodi Aadu goats, having a white coat, compared to Kanni Aadu goats with black-colored coats. It is a proven fact that the color black absorbs more radiant heat than a lighter/white color [26]. Hence, this could also be a first-time report to prove this fact using such a holistic approach wherein most of the observed variables are in favor of the white-colored coat of Kodi Aadu goats. Having said this, Stuart-Fox et al. [27] also discussed the thermo-protective role of increased melanization, especially in those animals reared in humid and hot environments. The authors stated that melanin may play the protective role of hydroregulation along with UV protection in such cases. This could therefore explain the increased DEG count and also relatively higher up-regulation of most genes and associated pathways in Kanni Aadu goats compared to Kodi Aadu goats.

Among the top 20 DEGs, the *LOC102176710*, predicted to be coding for hemoglobin fetal subunit beta, is the is top up-regulated gene in Kanni Aadu goat skin, while a similar gene, *LOC102182615*, predicted to be coding for hemoglobin subunit beta-like, is the most down-regulated gene in Kodi Aadu. The increased expression of this gene in Kanni Aadu goats could reflect the increased oxygen demand of the skin tissue due to the deleterious impact of heat stress at the cellular level. At the same time, the extreme down-regulation of the same gene in Kodi Aadu goats can project its supreme tolerance ability. Another set of vital pathways associated with hemoglobin production (HbF) is the eukaryotic initiation factor 2αP (eIF2αP)-activating transcription factor 4 (ATF4) pathway and integrated stress response (ISR) pathway. Reports have stated that eIF2αP stress signaling has been associated with the regulation of fetal hemoglobin production (HbF) [28]. Furthermore, the ISR, a common adaptive pathway reported in eukaryotic cells, is activated during stress and thereby aids to restore cellular homeostasis [29]. In our study, a number of genes involved in this pathway were found to be significantly up-regulated in Kanni Aadu goats, while the same were significantly down-regulated in Kodi Aadu goats. Though these genes have been primarily studied in erythrocytes, their influence and role in the skin during heat stress would be a novel finding.

The expression profiles of the molecular chaperones reflect the status of heat stress at cellular levels. It was interesting to observe the transcriptome profile of some of the classical heat-stress-associated and molecular-chaperones-associated genes in the skin tissue. The majority of the classical HSPs differentially expressed in this study were significantly up-regulated in Kanni Aadu skin during heat stress, while in Kodi Aadu goats, they were significantly down-regulated. This again points towards the excellent thermal tolerance ability of Kodi Aadu goats, which probably experience a subthreshold level of heat stress despite being subjected to a THI of 97.76. Such a mechanism of significantly lower expression of the *HSP70* gene during extreme heat stress was reported in Salem Black goats [21]. Therefore, this could be a unique trait exhibited by indigenous goats, possessing supreme thermo-tolerance. Along with the HSPs, ubiquilin is also a vital molecular chaperone that plays a critical role in the clearance of misfolded and aggregated proteins. Among the several ubiquilins, *UBQLN2* has been reported to form a network with HSP70 while clearing the aggregated proteins [30]. These genes, however, have not been explored well in livestock. Therefore, based on the results of the current study, for the first time ever, the association of *UBQLN2* and *HSP70.1* during heat stress can be substantiated. Likewise, the functionality of *UBQLN3* gene in caprine skin during heat stress may also be explored, as this gene was significantly down-regulated in Kanni Aadu goats.

Further, the gene ontology and KEGG analysis also depicts an evident effect of heat stress in caprine skin. It may be noted at this juncture that the two breeds exhibited a certain degree of similarity when it came to responding to heat stress. The GO term NADH dehydrogenase (ubiquinone) activity (MF) was common for both the breeds and is reported to be involved with the NADH:ubiquinone oxidoreductase (complex I). The complex I is the most thermolabile protein complex of oxidative phosphorylation, which breaks down during heat stress [31]. Apart from the above-stated MF, oxygen transporter activity was significantly enriched in Kanni Aadu goats and structural molecule activity in Kodi Aadu goats. Therefore, it is also noteworthy to observe the differences exhibited by Kanni Aadu and Kodi Aadu goats although they are both being indigenous breeds with their breeding tract from the same South Indian state.

Furthermore, the enrichment of the intermediate filament (CC) and keratin filament (CC), could be associated with the hair characteristics (fiber diameter and staple length) and skin epithelial height of Kodi Aadu goats. These traits were found beneficial for an animal to efficiently adapt to heat stress. Therefore, an evident alteration at the cellular level in the caprine skin tissue was identified due to heat stress. Apart from being a first-time report in goats, this also has novelty in depicting how two goat breeds, despite being indigenous, have some contrasting mechanisms in response to heat stress.

Apart from the DNA sequence, genetic information is also encoded by epigenetic modifications of the DNA that include DNA methylation and histone modification. These “epigenetic markers” influence gene expression by altering the chromatin structure [32]. Environmental stressors, both biotic and abiotic, can trigger these epigenetic patterns in an individual, which can accumulate over time, leading to a persistent alteration [32]. Therefore, epigenetics could constitute a main link between changing environmental conditions and animals’ response to maintain homeostasis [33]. Heat stress is one of the major consequences of environmental stress; however, minimal studies have been conducted to assess the changes occurring at the epigenetic level due to heat stress in livestock. Therefore, as a first-time approach, the skin epigenetic profiles of goats were assessed upon exposure to heat stress.

The variations in the overall genomic DNA methylation, though representing only a very marginal amount, should not be overlooked, as they represent the percentage across the entire caprine gene. Additionally, the high numbers of DMRs in both the breeds certainly reflect the impact of heat stress at the DNA methylation level in Kanni Aadu and Kodi Aadu skin. 

The classification of the majority of the CpG DMRs under the COG category “Signal transduction mechanism” reveals the cellular process and signaling occurring due to the deleterious impact of heat stress on the goats. Substantiating the COG distribution, most of the GO terms were enriched in both Kanni Aadu and Kodi Aadu goats, reflecting that a number of cell signaling and response pathways were enriched as a consequence of heat stress in caprine skin. Further, the different GO terms (generally reflecting the cell signaling and response mechanism) enriched in the two breeds also highlight the breed-specific heat-stress-response mechanism. Additionally, some of the GO terms identified in the present study were also reported by Del Corvo et al. [33], who led a study in Nellore and Angus cattle to assess their PBMC epigenomics responses to heat stress.

Linking the skin transcriptomics and epigenetics profile was another novelty of this study. It is noteworthy to observe the contrasting differential methylation and differential expression trends exhibited by Kanni Aadu and Kodi Aadu goats. Additionally, this further unraveled some novel findings regarding the impact of heat stress at the cellular and molecular levels in caprine skin tissue in addition to understanding the contrasting responses exhibited by the two goat breeds. It was noteworthy to identify some of the classical hair-follicle-, hair-growth-, and pigmentation-associated genes such as *KRT1*, *KRT10*, *KRT23*, *KRT35*, *COL1A1*, *COL1A2*, *COL2A1*, *COL3A1*, *FOS*, *FOSL2*, *KLF10*, *KLF13*, and *KLF17*. While most of these genes were up-regulated in Kanni Aadu goats, many were down-regulated in Kodi Aadu goats. Apart from contributing to the fiber strength, color, and skin properties, some of these genes are also associated with hair length [34,35]. This could substantiate the molecular mechanism during heat stress that aids the expression of short and thick hair fibers in Kodi Aadu goats, thereby enhancing the breed’s adaptability to thermal stress. Likewise, the differences in the differential expression profile of the heat-shock-protein- (*HSF1*, *HSF2BP*, *HSP70.1*, *HSPA8*, *HSPB1*, *HSPB3*, and *HSPB8*) and immune response (*IL2*, *IL4*, *IL10*, *IL1B*, and *IL15*)-related genes also indicate impaired adaptive responses in Kanni Aadu goats. This alteration at the molecular level could also have an influence on the higher number of significantly altered skin microbial genera (LEfSe analysis) in the heat-stressed Kanni Aadu goats compared to Kodi Aadu. 

## 4. Materials and Methods

### 4.1. Location of the Study

The experiment was conducted in a state-of-the-art climate chamber facility at the Centre for Climate Resilient Animal Adaptation Studies (CCRAAS), ICAR-National Institute of Animal Nutrition and Physiology (NIANP), experimental livestock farm, Bengaluru, India. Due approval from the ethical committee was obtained prior to the study, thereby permitting to subject the animals to heat stress, followed by slaughter (CPCSEA/2/2017).

### 4.2. Animal Details

A total of 24 female goats, aged between 9–12 months, were used for the study and were comprised of two breeds, namely Kanni Aadu (*n* = 12) and Kodi Aadu (*n* = 12). Both the indigenous goat breeds have their breeding tract from the South Indian state, Tamil Nadu, whence they were procured for the study. Care was taken to ensure heterozygosity among the population during procurement. Furthermore, all the selected animals were apparently healthy and with no visible health issues. While both the goat breeds are known for their hardy nature, they exhibit distinctive breed differences. Kanni Aadu goats are predominantly black-coated, with typical white or reddish-brown stripes on either side of the face, underbelly, and inner sides of the legs. Kodi Aadu goats have white-colored coats with black and brown patches that are scattered over the coat. After procurement, the animals were transported to ICAR-NIANP, Bengaluru, and were acclimatized for 45 days prior to the experiment.

### 4.3. Experimental Design

A 45-day experiment was conducted in the controlled-climate chamber of CCRAAS, ICAR-NIANP, between May and June 2020. The animals were randomly allocated into four groups of six animals each: KAC (*n* = 6; Kanni Aadu control), KAH (*n* = 6; Kanni Aadu heat stress), KOC (*n* = 6; Kodi Aadu control), and KOH (*n* = 6; Kodi Aadu heat stress). The animals were stall-fed with a standard 60:40 roughage–concentrate diet that is usually provided for all goats reared at the farm. All animals were also given ad libitum access to feed and water, individually. The control group animals, both KAC and KOC, were housed in the thermo-neutral zone (TNZ) chambers with a constant temperature from 23–24 °C during the experimental time (10:00 a.m. to 04:00 p.m.). The heat-stress groups, both KAH and KOH, were housed in the heating chamber with a simulated heat-stress model between 10:00 a.m. to 4:00 p.m. throughout the 45 days (Appendix A). All cardinal weather parameters were recorded twice every day during the entire duration of the study using a thermo-hygrometer (Vaishno Instruments, Telangana, India).

The temperature humidity index (*THI*) was calculated (morning and afternoon) following the Mc Dowell [12] equation, as given below:(1)$$THI=0.72\left(T_{db}+T_{wb}\right)+40.6$$
where *T_db_* = dry bulb temperature; *T_wb_* = wet bulb temperature. 

At the end of the study, on the 45th day, all the animals were slaughtered following all hygienic processes at the slaughter house at the Experimental Livestock Unit, ICAR-National Institute of Animal Nutrition and Physiology, Bengaluru.

### 4.4. Hair Fiber Analysis

To assess the hair characteristics, approximately 5 g of hair samples were collected from the rump region of goats at the 45th day of the experiment using a trimmer. The hair characteristics primarily analyzed were fiber diameter and staple length. All the fiber analyses were performed at ICAR-Central Sheep and Wool Research Institute, Rajasthan, India. The fiber diameter was measured according to IS 744 standard test method using a computerized microscope (RxLr-4, Radical Fiber plus) enabled with image-processing software (Radical fiber plus). Moreover, 300 fibers were averaged to arrive at the average fiber diameter.

Staple length is the length of a staple obtained by measuring it in an unstretched condition. The staple length was measured as per the IS 6653 standard method. The staple tip-to-tip distance was measured using a steel ruler in centimeters. Ten measurements were averaged to arrive at the staple length.

### 4.5. Hair Cortisol Estimation

Approximately 5g of hair was collected on the 45th day of the experiment and was processed for glucocorticoid extraction using the protocol earlier described by Broin et al. [16] with mild modification. The extracts were subjected to ELISA using Puregene Goat Cortisol ELISA kit (analytical sensitivity: 0.22 ng/mL; intra-assay and inter-assay coefficient of variations were <8% and <10%, respectively), following the standard manufacturer’s protocol. Further, the cortisol concentration was estimated using a micro plate reader (Thermo Scientific, Vantaa, Finland).

### 4.6. Hair Follicle qPCR Analysis

Hair follicles were collected from the tails of goats using a plucker on the 45th day of the study. On plucking the hair, the follicles were immediately dipped in DEPC-treated water to prevent RNAse activity. The hair samples were finely cut near the follicles and placed directly into 2 mL Eppendorf tubes. The collected follicles were immediately processed for RNA extraction using a modified TRI reagent protocol. A non-reacting metal ball was placed in the tubes containing the hair samples. The total RNA was isolated using a TRI reagent (Sigma Aldrich, St. Louis, MO, USA) and treated with DNase I (Sigma Aldrich) to prevent genomic DNA contamination, following the manufacturers’ guidelines. The quality and quantity of RNA were estimated using spectrophotometer (NanoDrop^TM^ 2000 C). Lastly, the Maxima first-strand cDNA synthesis kit (Thermo Scientific, Vilnius, Lithuania) was used to reverse transcribe the RNA samples into cDNA, which were then stored at −80 °C until further use.

Primers to amplify a short fragment of the target genes; heat-shock protein 70 (*HSP70*), *HSP90,* and *HSP110* and endogenous reference genes; hypoxanthine phosphoribosyl transferase (*HPRT*); and glyceraldehyde 3-phosphate dehydrogenase (*GAPDH*) were selected from the previously published literature [9]. Quantitative PCR (qPCR) was performed using SYBR green chemistry (Maxima SYBR green qPCR master mix, Fermentas, Waltham, MA, USA) by adopting a similar methodology as described by Archana et al. [8]. The relative expression of the targeted genes was assessed adopting the 2^−ΔΔCT^ method [36] using geometric means of *HPRT* and *GAPDH* genes.

### 4.7. Sweating Rate and Active Sweat Gland Estimation

The sweating rate of goats was recorded at fortnightly intervals following the methodologies described by Schleger and Turner [37] (in cattle) and Maurya et al. [23] (in sheep) with mild modification. The sweating rate (*SR*) was calculated as below, and the unit of measurement is depicted as gm^−2^ h^−1^.
(2)$$SR=\frac{38,400}{time\;in\;seconds}$$

The active sweat glands were estimated using a modified iodine paper technique (modified protocol of Gagnon et al. [38]). Two days prior to the recording, prefilter paper (Merck Millipore, Bangalore, India) of known size was placed in a sealed container. The container was filled with iodine crystals at the bottom, and the prefilter paper was suspended such that it did not come into direct contact with the iodine crystals. Further, the container was covered with aluminum foil to avoid iodine crystals being exposed to light. After 2 days, the prefilter paper turned dark brown in color, depicting its saturation with iodine and indicating that it could be used for recording. On the day of the recording, the shaved area of the goat skin (7th–9th rib, midway from the dorsal vertebra and ventral sacrum) was first wiped using tissue paper to avoid any external contamination. The iodine-impregnated prefilter paper was pressed firmly on the goat skin, and to ensure even distribution and avoid external moisture from the handler’s hand, glass slides were placed over the prefilter paper. The iodine-impregnated prefilter paper was placed over the skin for 5 min. Sweat excreted from the active sweat glands reacted with the iodine impregnated paper to form dark dots, which were easily and grossly identified grossly. The filter paper was then scanned immediately using a commercial scanner for subsequent analysis. The total number of blue dots was counted to reflect the total number of active sweat glands. Further, this number of active sweat glands was divided by the surface area of the prefilter paper to represent the number of sweat glands per square centimeter.

### 4.8. Skin Histology

Skin-tissue samples covering the area around 7th–9th rib, midway from the dorsal vertebra and ventral sacrum, were collected aseptically during slaughter from all the goats in 10% buffered formal saline solution. All the samples after fixation were processed for sectioning and staining wherein 4 micron sections were obtained and stained with Hematoxylin and Eosin dye. Histomorphometric analysis was performed on all the stained sections using a Nikon Trinocular Microscope Model Ci-L (Tokyo, Japan) with NIS Elements image-analysis software to measure the skin epithelial height and also count the number of sweat glands.

### 4.9. Infrared Thermography of Caprine Skin

The skin-surface temperature was recorded at fortnightly intervals with four collections at 08.00 h, 14.00 h, 20.00 h, and 02.00 h. Infrared thermography (IRT) was performed using a handheld digital thermal imaging infrared camera (TI200–9 Hz, image resolution; 200 × 150 pixels, Fluke; Fluke Corporation, Everett, WA, USA). The IRT was recorded for five regions, i.e., the eye, forehead, flank, and back and front legs from a distance of approximately 1.0 m from the animal, with the camera lens held perpendicularly to each body location. For each targeted body part, at least two to three infrared images were taken. The images were interpreted using Fluke Smartviewer™ 4.0 (Fluke Corporation, EUA) software. 

### 4.10. Skin 16S rRNA V3-V4 Metagenomics 

The skin microbial profiles of goats were assessed using the 16s V3-V4 rRNA sequencing approach. Approximately 5 × 5 cm^2^ area of shaved skin tissue (covering the area around 7th–9th rib, midway from the dorsal vertebra and ventral sacrum) was collected aseptically during slaughter and stored in normal saline for further analysis. The bacterial cells from the skin-tissue samples were extracted as per the protocol by Rajesh et al. [39]. The DNA was extracted from the separated cells using the DNeasy PowerSoil Kit (Catalog: 12888-100, Qiagen, Hilden, Germany), by following the manufacturer’s protocol. The DNA was finally eluted in 25 µL of Solution C6 that was provided by the manufacturer. The DNA samples were quality-checked using a NanoDrop™ 2000 Spectrophotometer (ND2000, Thermo Scientific) to determine the purity. The DNA samples were then processed for library preparation targeting the V3–V4 hyper variable regions of the 16S rRNA gene, followed by use of the Illumina MiSeq sequencing platform at M/s Clevergene, Bangalore, India. 

The sequence data generated using Illumina MiSeq were checked for their quality using FastQC v0.12.1 version and MultiQC software. All the samples passed the QC threshold (Q20 > 95%) and were used for further data analysis. The raw reads were trimmed (20 bp) from 5′ end to remove the degenerate primers. The trimmed reads were processed to remove adapter sequences and low-quality bases using Trimgalore. The QC-passed reads were imported into mothur, and the pairs were aligned with each other to form contigs that were then further assessed to delete ambiguous base calls and merge duplicates. Further, the UCHIME algorithm was used to flag contigs with chimeric regions. The filtered contigs were processed and classified into taxonomical outlines based on the GREENGENES v.13.8-99 database. The contigs were then clustered into operational taxonomic units (OTUs). After the classification, OTU abundance was estimated. For the functional analysis, PICRUSt was used to predict gene family abundance. The 16s RNA copy numbers were normalized by PICRUSt’s precalculated files. The metagenomes were predicted using predict_metagenomes.py script. OTU contributions for the particular functions were estimated by metagenome_contributions.py script.

### 4.11. Skin Transcriptomics Analysis

A portion of the skin tissue covering the thoracic region of the goat was collected aseptically during slaughter and stored in RNA later at −80 °C until further extraction. RNA was extracted from the skin tissue using the RNA Miniprep (Cat No.74104), following the manufacturers protocol with slight modifications.

The RNA quality assessment was performed using RNA ScreenTape System and assessed using Agilent 4150 TapeStation instrument (Agilent Technologies, Inc, Santa Clara, CA, USA). Additionally, RNA concentration was determined using a Qubit^®^ 3.0 Fluorometer (Catalog: Q33216, Thermo Fisher Scientific, Waltham, MA, USA) and using the Qubit™ RNA BR Assay Kit (Catalog: Q32853, Thermo Fisher Scientific). All the samples that passed the quality check were then processed for mRNA enrichment and library preparation. Then, 500 ng of total RNA was used to enrich the mRNA using NEBNext Poly (A) mRNA magnetic isolation module (Catalog: E7490, New England Biolabs, Ipswich, MA, USA) by following the manufacturers’ protocol. The enriched mRNAs were further taken for the library preparation using the NEBNext^®^ Ultra™ II RNA Library Prep Kit for Illumina (Catalog: E7775S, New England Biolabs). The library concentration was determined in a Qubit.3 Fluorometer (Catalog: Q33216, Life technologies, Carlsbad, CA, USA) using The Qubit dsDNA HS (High Sensitivity) Assay Kit (Catalog: Q32854, Thermo Fisher Scientific). The library quality assessment was performed using Agilent D1000 Screen Tape System (Catalog: 5067-5582, Agilent) in a 4150 Tape Station System (Catalog: G2992AA, Agilent). All the libraries that passed the quality check were then set for sequencing in the Illumina HiSeq sequencing platform at M/s Clevergene, Bangalore, India.

The sequence data that were generated were checked for quality using FastQC and MultiQC software. All the samples that passed the QC threshold (Q20 > 95%) were processed to remove adapter sequences and low-quality bases using fastp. Further, the QC passed reads were mapped onto the indexed goat (*Capra hircus*) reference genome (assembly: ARS1, GCF_001704415.1) using STAR v2 aligner. For differential expression analysis, the biological replicates were grouped as control and treatment (for both the breeds separately). Differential expression analysis was carried out using the edgeR package after normalizing the data based on the trimmed mean of M (TMM) values. Genes with absolute log2 fold change ≥ 1 and *p*-value ≤ 0.05 were considered significant. The UpSetR R package was used to generate plots showing overlapping significant genes between conditions. The expression profile of differentially expressed genes across the samples was presented in volcano plots. Further, the genes that showed significant differential expression were used for gene ontology (GO) and pathway enrichment analysis.

Enrichment analysis for biological process (BP), Molecular function (MF), Cellular component (CC), and KEGG pathway was performed using DAVID. Gene ontology (GO) and KEGG pathways with multiple test-adjusted *p*-value ≤ 0.05 were considered significant. To visualize the GO enrichment results, the GOplot R package was used.

### 4.12. Skin Bisulfite Sequencing

The epigenetic profile of caprine tissue was assessed by whole-genome bisulfite sequencing. Skin tissue was aseptically collected from the thorax of goats during slaughter and stored in phosphate-buffered saline (PBS) at −80 °C until further extraction. Genomic DNA was extracted from the collected skin tissue samples using the DNeasy PowerSoil Kit (Catalog: 12888-100, Qiagen) following the manufacturers. The DNA samples were quality-checked using Qubit 3.0, and their integrity was assessed using 0.8% agarose gel electrophoresis. Further, the DNA concentration of the extracted samples was determined by using Qubit™ dsDNA BR Assay Kit (Catalog: Q32853, Thermo Fisher Scientific, and the readings were taken in a Qubit 3.0 Fluorometer (Thermo Fisher Scientific).

All the samples that passed the QC were processed for library preparation for whole-genome bisulfite sequencing (WGBS) using Pico Methyl-Seq Library Prep Kit (Catalog: D5456, Zymo Research). The bisulfite converted DNA was amplified by 6 cycles of PCR with the addition of Library Amp Master Mix and Index primers to facilitate multiplexing while sequencing. The amplified product was then purified using Zymo spin-IC column, and the final library was eluted in 20 μL of DNA elution buffer. Further, the concentration and quality of the libraries were assessed using the Qubit.3 Fluorometer (Catalog: Q33216, Life technologies) and Agilent D1000 ScreenTape System (Catalog: 5067-5582, Agilent), respectively. All the samples passed the library QC and were then subjected to sequencing using Illumina HiSeq at M/s Clevergene, Bangalore, India.

The sequence data were processed to remove the adapter content and were trimmed at 5′ end using fastp. The trimmed reads were mapped to the reference goat genome GCF_001704415.1_ARS1, and duplicate reads were removed using the bismark tool. Methylation levels were estimated using the methlKit R bioconductor package. The methylation preferences were plotted using the WebLoGo. Additionally, the hypermethylated CGI (CG islands) regions (hypermethylation CGI definition: methylation level over 0.7 at CpG) were also assessed. The hypermethylated CGIs were annotated using HOMER annotatePeaks.pl script. The annotations were plotted using the plotly R package. The methlKit was used to perform differential methylation analysis. The differentially methylated regions (DMRs) were considered significantly methylated if the q value was <0.05, and the minimum differential methylation level was 0.3 (CpG) and 0.2 (CpH, CHH). The differentially methylated regions were annotated against the goat reference genome using HOMER annotatePeaks.pl script. The closest genes of the DMRs were used for the GO and KEGG pathway over representation analysis using DAVID. COG annotations were obtained by performing BLAST against the NCBI COG 2020 database. The cluster of orthologous groups (COGs) annotations of the DMR genes was plotted using the ggplot2 R package.

### 4.13. Statiscial Analysis

The data obtained for hair fiber, hair cortisol, sweating rate, active sweat gland estimation, histomorphometry, IRT, and qPCR analysis were analyzed using the General Linear Model (GLM) of SPSS V.21 using breed and group as fixed effects. The correlation coefficient between breed, group, and THI with hair characteristics, hair cortisol, hair follicle qPCR, sweating, skin histometry, and skin-surface IRT were assessed by Pearson’s correlation coefficient test using SPSS V.21. Lastly, suitable statistical analysis wherever necessary was performed for the NGS approaches, which are listed in their respective sections. The results are represented as mean ± standard error (SE), with the significance level set at *p* < 0.05.

## 5. Conclusions

Our study highlights some novel findings through this innovative approach by assessing the impact of heat stress on caprine skin tissue. Not only did heat stress have a significant impact on this vital tissue, but it also explains how two indigenous goat breeds exhibited contrasting responses to the same stress. At the phenotypic level, most of the hair- and skin-associated variables pointed towards the better resilience of Kodi Aadu to heat stress. The skin microbiota was also observed to be significantly altered due to heat stress, indicating a probable threat to the integrity of the animal’s protective barrier. At the molecular level, it was noteworthy to observe the evident differences in the transcript expression profile and epigenetic alterations between the two goat breeds. This study revealed a significant alteration in some of the well-established coat characteristics and adaptation- and immune-response-related genes. Thus, based on the findings from this study, we found the skin tissue to be a potential medium to assess climate resilience in goats. Additionally, though Kanni Aadu and Kodi Aadu goats possess excellent climate-resilience potential, Kodi Aadu goats stand out as more tolerant to heat stress.

## Figures and Tables

**Figure 1 ijms-24-10319-f001:**
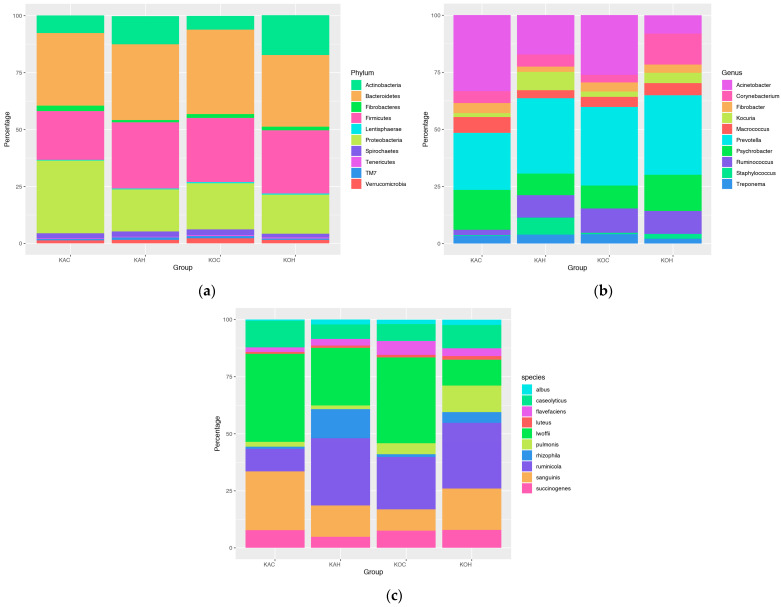
The relative abundance of 10 most abundant skin microbial phyla (**a**), genera (**b**), and species (**c**) in Kanni Aadu and Kodi Aadu goat breeds subjected to heat stress with their respective control groups (KAC, Kanni Aadu control; KAH, Kanni Aadu heat stress; KOC, Kodi Aadu control; KOH, Kodi Aadu heat stress). The relative abundance of different microbial species varied within both the goat breeds during heat stress.

**Figure 2 ijms-24-10319-f002:**
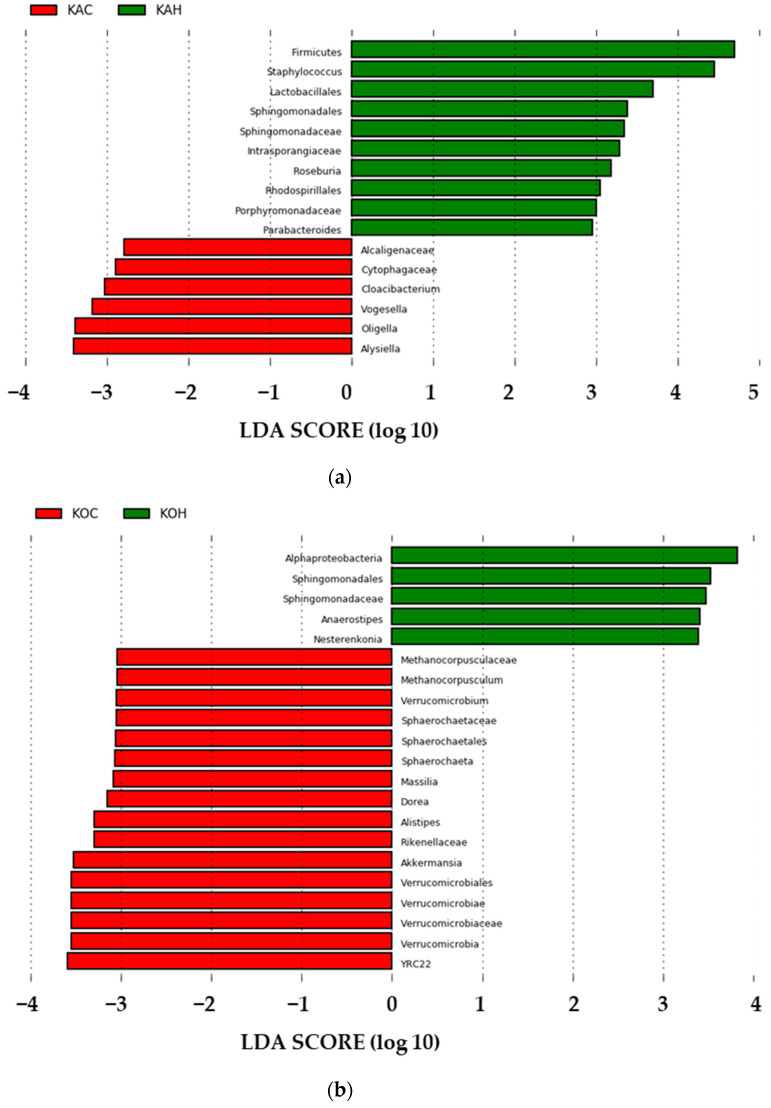
LEfSe analysis depicting the significant microbial genera on comparing the KAC vs. KAH (**a**) and KOC vs. KOH (**b**).

**Figure 3 ijms-24-10319-f003:**
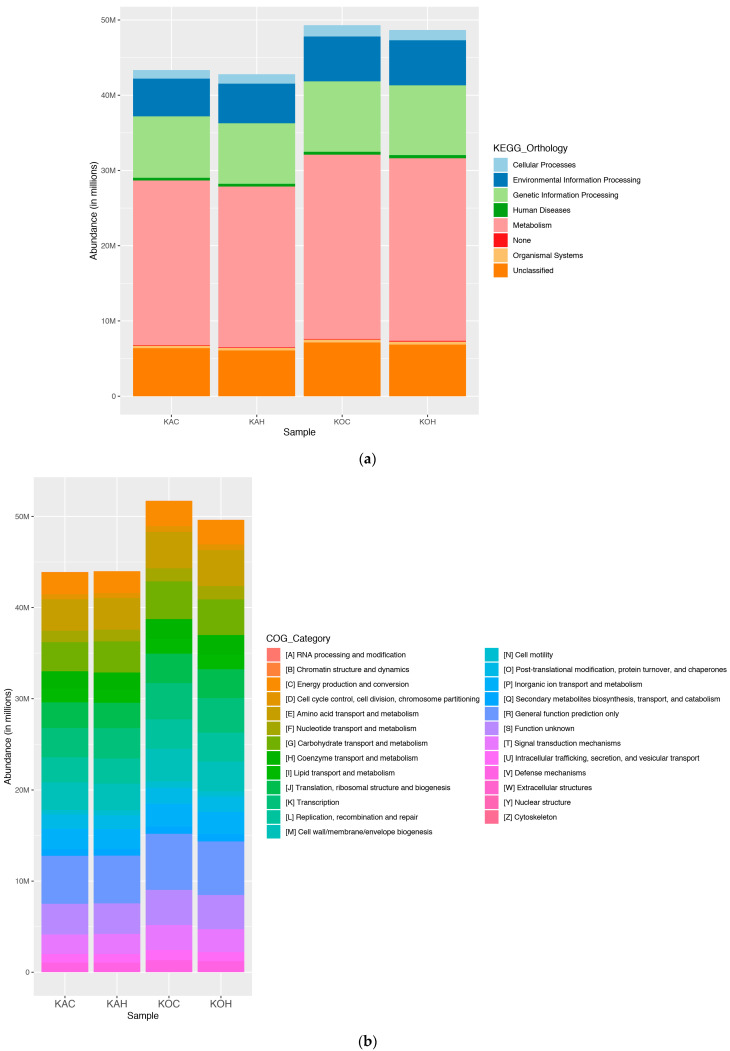
Functional prediction of the skin microbiome using PICRUSt analysis at KEGG-1 (**a**) and clusters of orthologous groups of proteins (COGs) (**b**).

**Figure 4 ijms-24-10319-f004:**
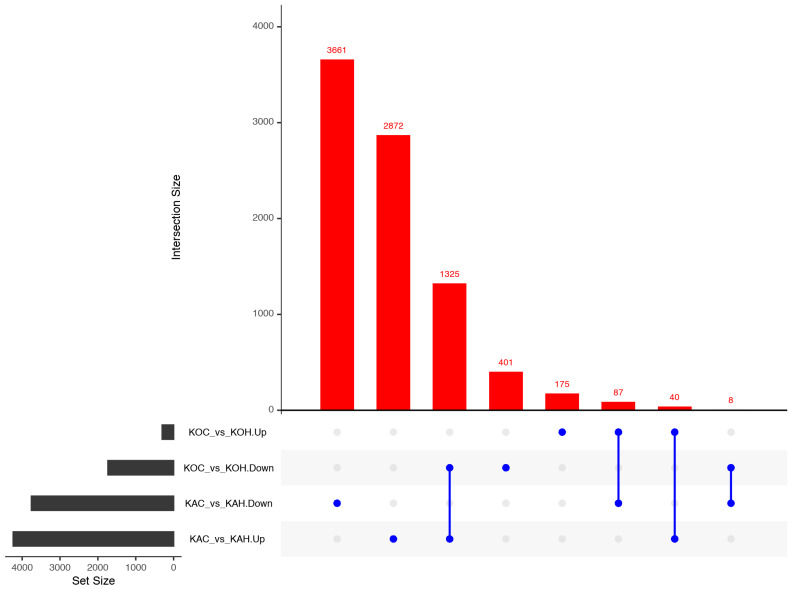
UpSetR plot representing distribution of up- and down-regulated genes in KAC_vs_KAH and KOC_vs_KOH comparisons.

**Figure 5 ijms-24-10319-f005:**
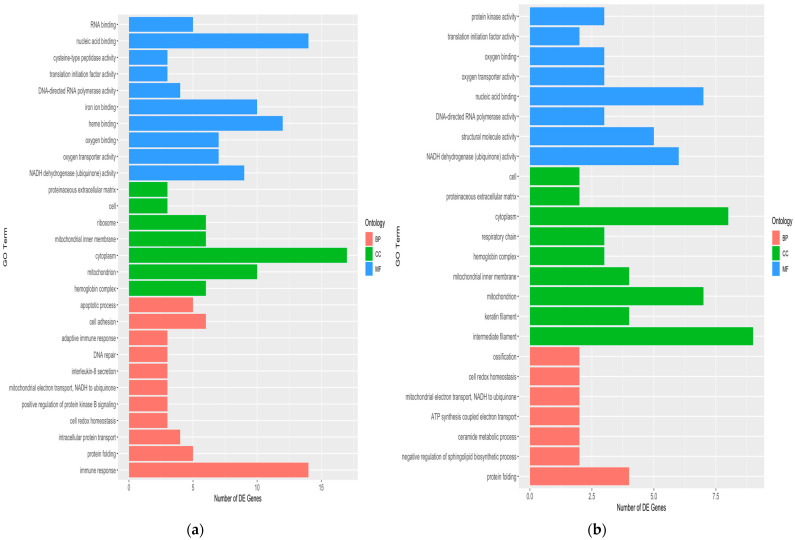
An overview of the gene ontology terms in KAC_vs_KAH (**a**) and KOC_vs_KOH (**b**) based on genes significantly expressed in skin tissue. MF, molecular function; CC, cellular component; BP, biological process.

**Figure 6 ijms-24-10319-f006:**
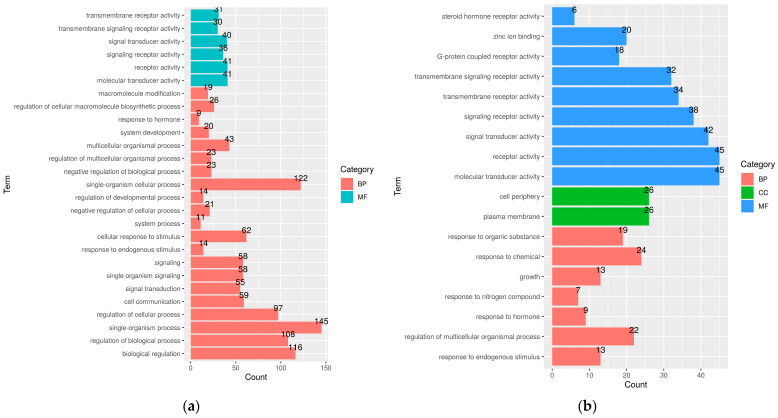
An overview of the gene ontology terms in KAC_vs_KAH (**a**) and KOC_vs_KOH (**b**) based on DMGs significantly methylated in skin tissue. MF, molecular function; CC, cellular component; BP, biological process.

**Figure 7 ijms-24-10319-f007:**
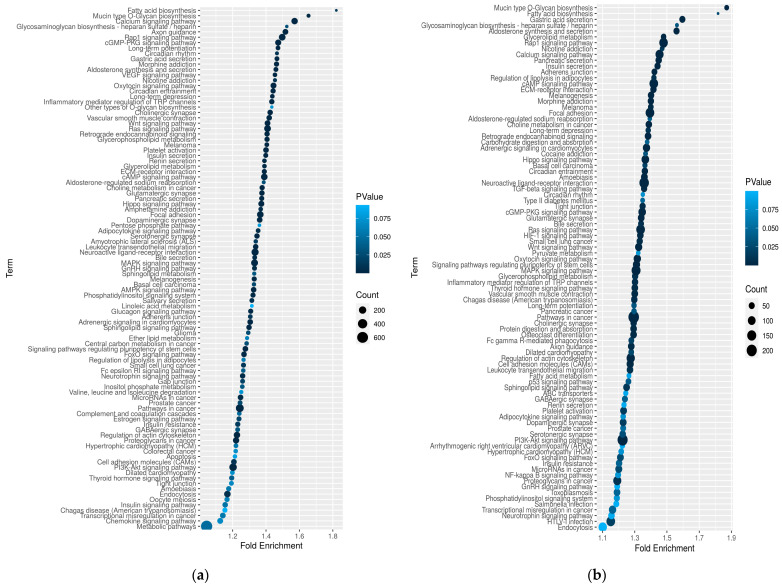
An overview of the KEGG pathways enriched in KAC_vs_KAH (**a**) and KOC_vs_KOH (**b**) based on skin methylation profile.

**Figure 8 ijms-24-10319-f008:**
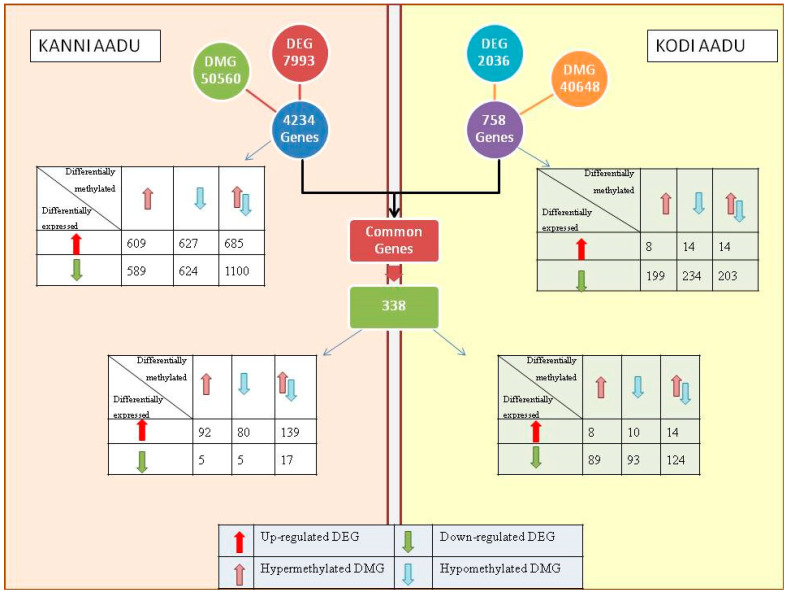
Pictorial depiction of linking the differentially methylated and expressed genes in Kanni Aadu and Kodi Aadu goats as a consequence of heat stress.

**Table 1 ijms-24-10319-t001:** Results for the phenotypic and genotypic variables recorded in Kanni Aadu and Kodi Aadu goats on exposure to heat stress.

	Variable	Kanni Aadu		Kodi Aadu	
		KAC	KAH	KOC	KOH
Hair characteristics	Fiber diameter (µm)	68.00 ± 2.43 ^a^	64.33 ± 1.49 ^a^	82.93 ± 3.18 ^b^	84.24 ± 3.97 ^b^
		*p*-value: 0.00			
	Staple length (cm)	3.77 ± 0.03 ^b^	3.53 ± 0.12 ^ab^	3.72 ± 0.27 ^b^	3.11 ± 0.17 ^a^
		*p*-value: 0.05			
Hair cortisol	Hair cortisol (ng/mL)	6.89 ± 0.16 ^a^	6.67 ± 0.16 ^a^	6.92 ± 0.22 ^a^	6.62 ± 0.16 ^a^
		*p*-value: 0.54			
Hair follicle qPCR	*HSP70*	1.00 ± 0.08 ^a^	0.99 ± 0.12 ^a^	1.00 ± 0.13 ^a^	0.52 ± 0.19 ^b^
		*p*-value: 0.89		*p*-value: 0.02	
	*HSP90*	1.00 ± 0.11 ^a^	0.63 ± 0.07 ^b^	1.00 ± 0.01 ^a^	0.26 ± 0.10 ^b^
		*p*-value: 0.01		*p*-value: 0.00	
	*HSP110*	1.00 ± 0.09 ^a^	0.98 ± 0.06 ^a^	1.00 ± 0.15 ^a^	0.68 ± 0.07 ^b^
		*p*-value: 0.81		*p*-value: 0.05	
Sweating	Sweating rate (g/m^2^/h)	0.00 ± 0.00 ^a^	2.80 ± 0.37 ^b^	0.00 ± 0.00 ^a^	2.82 ± 0.37 ^b^
		*p*-value: 0.00			
	Active sweat gland measurement (No of gland/cm^2^)	0.00 ± 0.00 ^a^	0.03 ± 0.01 ^b^	0.00 ± 0.00 ^a^	0.06 ± 0.01 ^b^
		*p*-value: 0.00			
Skin histometry	Epithelial height (µm)	15.62 ± 0.23 ^a^	16.84 ± 0.18 ^b^	18.15 ± 0.49 ^c^	22.94 ± 0.60 ^d^
		*p*-value: 0.00			
	No. of sweat glands/cm^2^	421.67 ± 3.53 ^a^	422.75 ± 1.83 ^ab^	429.33 ± 2.14 ^bc^	431.42 ± 1.97 ^c^
		*p*-value: 0.00			
Skin-surface infra-red thermal imaging	Eye (°C)	36.93 ± 0.11 ^a^	40.96 ± 0.12 ^b^	36.88 ± 0.15 ^a^	41.14 ± 0.09 ^b^
		*p*-value: 0.00			
	Forehead (°C)	29.39 ± 0.10 ^a^	40.44 ± 0.47 ^b^	29.83 ± 0.24 ^a^	40.28 ± 0.13 ^b^
		*p*-value: 0.00			
	Flank (°C)	30.68 ± 0.26 ^a^	40.78 ± 0.22 ^b^	31.34 ± 0.27 ^a^	40.43 ± 0.13 ^b^
		*p*-value: 0.00			
	Back (°C)	28.73 ± 0.47 ^a^	40.41 ± 0.16 ^c^	29.63 ± 0.24 ^b^	39.83 ± 0.13 ^a^
		*p*-value: 0.00			
	Front leg (°C)	27.06 ± 0.34 ^a^	40.71 ± 0.25 ^b^	27.30 ± 0.13 ^a^	40.38 ± 0.20 ^b^
		*p*-value: 0.00			
Skin 16S rRNA V3-V4 metagenomics	Relative abundance of microbes at phylum level (%)				
	*Bacteroidetes*	31.86	33.23	37.10	31.38
	*Firmicutes*	21.40	29.00	28.13	27.76
	*Proteobacteria*	31.86	18.54	20.21	17.12
	*Actinobacteria*	7.67	12.44	6.02	17.48
	*Spirochaetes*	2.25	2.27	2.52	1.58
	*Fibrobacteres*	2.40	1.04	1.68	1.56
	*Verrucomicrobia*	1.26	1.52	2.30	1.49
	*TM7*	0.61	1.23	1.04	0.69

KAC, Kanni Aadu control; KAH, Kanni Aadu heat stress; KOC, Kodi Aadu control; KOH, Kodi Aadu heat stress. Means with different superscripts differ significantly; *p* < 0.05.

**Table 2 ijms-24-10319-t002:** Correlation coefficient between breed, group, and THI with hair characteristics, hair cortisol, hair follicle qPCR, sweating, skin histometry, and skin-surface IRT.

				Hair Characteristics		Hair Cortisol	Hair Follicle qPCR			Sweating		Skin Histometry		Skin-Surface Infra-Red Thermal Imaging				
	Breed	Group	THI	FD	SL	HCC	HSP70	HSP90	HSP110	SR	ASG	SEH	SG	Eye	FH	Back	Flank	L
**Breed**	1																	
**Group**	0.89 **	1																
**THI**	0.00	0.45 *	1															
**FD**	0.79 **	0.69 **	−0.05	1														
**SL**	−0.26	−0.43 *	−0.46 *	−0.27	1													
**HCC**	−0.01	−0.15	−0.31	0.23	0.01	1												
**HSP70**	0.84 **	0.90 **	0.34	0.68 **	−0.43 *	−0.05	1											
**HSP90**	0.74 **	0.93 **	0.60 **	0.58 **	−0.52 **	−0.25	0.91 **	1										
**HSP110**	0.86 **	0.91 **	0.31	0.76 **	−0.35	−0.13	0.91 **	0.91 **	1									
**SR**	0.00	0.44 *	0.99 **	−0.07	−0.44 *	−0.34	0.34	0.60 **	0.31	1								
**ASG**	0.20	0.55 **	0.82 **	0.23	−0.59 **	−0.25	0.50 *	0.72 **	0.48 *	0.83 **	1							
**SEH**	0.69 **	0.83 **	0.48 *	0.58 **	−0.45 *	−0.175	0.78 **	0.88 **	0.83 **	0.47 *	0.51 *	1						
**SG**	0.57 **	0.56 **	0.11	0.38	−0.30	−0.19	0.59 **	0.49 *	0.55 **	0.12	0.14	0.41 *	1					
**Eye**	0.02	0.46 *	0.99 **	−0.02	−0.45 *	−0.28	0.37	0.62 **	0.33	0.97 **	0.83 **	0.49 *	0.12	1				
**FH**	0.01	0.46 *	0.99 **	−0.04	−0.47 *	−0.34	0.34	0.60 **	0.30	0.98 **	0.83 **	0.49 *	0.1	0.99 **	1			
**B**	0.01	0.46 *	0.99 **	−0.02	−0.42 *	−0.28	0.33	0.59 **	0.29	0.98 **	0.87 **	0.47 *	0.09	0.99 **	0.99 **	1		
**Flank**	0.02	0.46 *	0.99 **	−0.02	−0.42 *	−0.29	0.34	0.59 **	0.31	0.98 **	0.82 **	0.48 *	0.06	0.99 **	0.99 **	0.997 **	1	
**L**	−0.00	0.44 *	0.997 **	−0.04	−0.45 *	−0.31	0.33	0.59 **	0.3	0.98 **	0.82 **	0.48 *	0.10	0.99 **	0.995 **	0.99 **	0.99 **	1

** Correlation is significant at the 0.01 level (2-tailed); * correlation is significant at the 0.05 level (2-tailed). THI, temperature humidity index; F, fiber diameter; SL, staple length; HCC, hair cortisol concentration; SR, sweating rate; ASG, active sweat gland number; SEH, skin epithelial height; SG, sweat gland number; FH, forehead; B, back; L, leg (foreleg).

**Table 3 ijms-24-10319-t003:** Results for the genotypic variables recorded in Kanni Aadu and Kodi Aadu goats on exposure to heat stress.

	Variable	Kanni Aadu (KAC vs. KAH)	Kodi Aadu (KOC vs. KOH)
Skin transcriptomics	DEGs	7993	2036
	Up-regulated DEGs	4237	302
	Down-regulated DEGs	3756	1734
Skin whole-genome bisulfite sequencing	DMR	50,560	40,648
	Hyper-methylated DMR	25,178	19,657
	Hypo-methylated DMR	25,382	20,991
	DMG	14,646	13,388
	Hyper-methylated DMG	7336	6507
	Hypo-methylated DMG	7310	6904

KAC, Kanni Aadu control; KAH, Kanni Aadu heat stress; KOC, Kodi Aadu control; KOH, Kodi Aadu heat stress.

**Table 4 ijms-24-10319-t004:** DEGs associated with some of the stress-associated pathways that were significantly expressed Kanni Aadu and Kodi Aadu skin.

KAC_vs_KAH		KOC_vs_KOH	
Gene	Log2FC	Gene	Log2FC
**EIF–ATF pathway**			
*EIF2A*	1.775		
		EIF2B1	−1.343
*EIF2B2*	1.398		
*EIF2B4*	2.026	EIF2B4	−1.337
*EIF2B5*	1.547	EIF2B5	−1.77
*EIF2S1*	1.702		
		EIF2S2	−1.416
*EIF2S3*	1.752	EIF2S3	−1.493
*ATF4*	2.369	ATF4	−1.875
*ATF5*	2.519	ATF5	−1.367
**Stress-associated molecular chaperones**			
*UBQLN2*	1.593	UBQLN2	−1.437
*UBQLN3*	−7.974		
*HSF1*	2.078		
*HSP70.1*	3.035	HSP70.1	−1.492
*HSP90AB1*	2.285		
*HSP90B1*	1.863	HSP90B1	−1.392
		HSBP1L1	−2.111
		HSPA13	−1.108
		HSPA14	−1.344
		HSPA4	−1.1
*HSPA5*	2.021	HSPA5	−1.591
*HSPA8*	2.098	HSPA8	−1.756
*HSPA9*	1.805		
*HSPB1*	4.01	HSPB1	−1.898
*HSPB3*	−2.07		
*HSPB6*	3.04	HSPB6	−1.665
*HSPB8*	2.799	HSPB8	−1.901
*HSPBP1*	1.771		
*HSPD1*	1.555		

KAC, Kanni Aadu control; KAH, Kanni Aadu heat stress; KOC, Kodi Aadu control; KOH, Kodi Aadu heat stress.

**Table 5 ijms-24-10319-t005:** Overview of some of the KEGG pathways obtained through transcriptomics analysis that were significantly altered due to heat stress in Kanni Aadu and Kodi Aadu goat skin due to heat stress.

KAC_vs_KAH			KOC_vs_KOH		
Description	No. of DEG	*p*-Value	Description	No. of DEG	*p*-Value
Metabolic pathways	499	0.00	Metabolic pathways	189	0.00
Neuroactive ligand–receptor interaction	124	0.00	Huntington’s disease	64	0.00
Ribosome	113	0.00	Oxidative phosphorylation	62	0.00
Huntington’s disease	109	0.00	Parkinson’s disease	61	0.00
Biosynthesis of antibiotics	105	0.00	Alzheimer’s disease	58	0.00
Endocytosis	97	0.02	Biosynthesis of antibiotics	51	0.00
Parkinson’s disease	93	0.00	Spliceosome	35	0.00
Oxidative phosphorylation	90	0.00	Endocytosis	34	0.03
Protein processing in endoplasmic reticulum	81	0.00	RNA transport	28	0.01
Spliceosome	77	0.00	Protein processing in endoplasmic reticulum	28	0.01
RNA transport	71	0.01	Carbon metabolism	24	0.00
Lysosome	54	0.02	Proteasome	23	0.00
Carbon metabolism	53	0.01	Lysosome	22	0.01
Epstein–Barr virus infection	48	0.05	Cardiac muscle contraction	20	0.00
Retrograde endocannabinoid signaling	44	0.05	Ribosome biogenesis in eukaryotes	18	0.00
Antigen processing and presentation	41	0.00	Biosynthesis of amino acids	16	0.00
Cardiac muscle contraction	41	0.00	Pyrimidine metabolism	16	0.04
GABAergic synapse	41	0.01	Glutathione metabolism	11	0.02
Proteasome	37	0.00	Citrate cycle (TCA cycle)	9	0.01
Biosynthesis of amino acids	36	0.01	Pyruvate metabolism	9	0.03
Bile secretion	36	0.02	RNA polymerase	8	0.02
Arginine and proline metabolism	29	0.00	Steroid biosynthesis	7	0.02
Fat digestion and absorption	27	0.00			
Glutathione metabolism	26	0.01			
Ether lipid metabolism	23	0.04			
Phototransduction	17	0.00			
alpha-Linolenic acid metabolism	16	0.02			
RNA polymerase	16	0.04			
Citrate cycle (TCA cycle)	16	0.05			

**Table 6 ijms-24-10319-t006:** Whole-genome DNA bisulfite sequencing data representing the read statistics and genomic methylation percent.

Group	Total Reads	QC Passed Reads	% Mapped Reads	% Methylated Cs	% Methylated CpG	% Methylated CHG	% Methylated CHH
**KAC**	49,128,009	48,056,118	50.35	5.97	85.54	3.34	11.12
**KAH**	54,655,288	53,458,502	52.66	5.90	85.71	3.22	11.07
**KOC**	56,831,837	55,694,741	52.21	5.69	84.84	3.40	11.75
**KOH**	48,747,713	47,743,401	51.62	5.61	85.03	3.32	11.65

KAC, Kanni Aadu control; KAH, Kanni Aadu heat stress; KOC, Kodi Aadu control; KOH, Kodi Aadu heat stress; Cs, cytosines; CpG, shorthand for 5’—C—phosphate—G—3’; CHG and CHH, where H is adenine (A), cytosine (C), or thymine (T).

## Data Availability

All the data that support the findings of this study are available in the paper and its Appendix A published online.

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
