# Peer review of "Novel Insights to Assess Climate Resilience in Goats Using a Holistic Approach of Skin-Based Advanced NGS Technologies"

_ijms, 2023, doi:10.3390/ijms241210319_

Round 1
Reviewer 1 Report
No of animals within the group is small - affecting the statistics
line 58-60 needs more profound explanation. Introduction part is very brief.
Initial health status of the goats?
List of primers?
Statistical analysis in M&M part is missing
Figs& Tables well prepared
The reviewer would like to see HE stained slides photos - skin histology
Aspects of heat stres affecting skin mircobiome very interesting
Fig. 5&6&7 must be improved - barely visible letters
No information about the slaughter details - ethical concerns
Reviewer 2 Report
Good afternoon. I thank the authors and the editorial board for the opportunity to review this article. The work provides relevant and important information, however, after reading I have a few questions.
1. Table 1 is presented in a really complicated format. I recommend to present it in another form.
2. Could you describe the conditions of animals in more detail. Was the health of the experimental animals monitored? Can you state that the animals were healthy throughout the experiment?
3. Can you state that all the animals were purebred? Were the animals tested for purebredness?
4. What temperature conditions did you use to provoke stress?
5. Why is there a different number of genes in Table 3, for the two breeds studied?
6. The following abbreviations - cs cpg chg chh - I recommend writing in a note to Table 5.
7. In addition, I want to advise to consider splitting the article into 2 parts, because there are so many results and the information will be perceived better in separate parts.
Author Response
We, the authors, highly appreciate the time and effort put in by the editor and reviewers in providing their feedback on our manuscript, and are grateful for the suggestions put forth by the reviewers for the improvement of the same. We tried our best to address and incorporate most of the suggestions wherever it was appropriate in the revised version of manuscript.
Query 1: Table 1 is presented in a really complicated format. I recommend to present it in another form.
Response: Thank you for your suggestion. We have split the table into two and hope these edited tables (Table 1 and Table 3 as per the revised manuscript) looks better now
Query 2: Could you describe the conditions of animals in more detail. Was the health of the experimental animals monitored? Can you state that the animals were healthy throughout the experiment?
Response: All the animals selected for the experiment were apparently healthy and did not exhibit any signs of ill-health. Furthermore, they were subjected to 45 days acclimatization period wherein the animals were dewormed and vaccinated. During the experimental period the animals were regularly monitored. Apart from exhibiting signs of distress due to the subjected treatment (observed only in heat stress treatment group wherein the animals either had increased respiration rate, rectal temperature, postural differences), the animals did not exhibit any signs of ill-health then too.
Query 3: Can you state that all the animals were purebred? Were the animals tested for purebredness?
Response: We appreciate your concern. All the animals selected for the study were purebreds (purebred Kanni Aadu and purebred Kodi Aadu goats). The animals were screened and visually inspected by a certified Veterinarian (with experience in animal breed identification). As goat farming in most parts of India is a small scale and extensively practiced system, there aren’t any specific breed registers for most cases. However, identification and selection of these animals were done upon considering a number of factors like visual breed characteristics, assessing the entire goat flock owned by a farmer, inquiring about the farming history of the farmer, and so on.
Query 4: What temperature conditions did you use to provoke stress?
Response: Thank you for your query. We used a simulated heat stress model to induce heat stress in the current experimental group. The detailed temperature humidity set at specific time intervals have been depicted in supplementary table S5
Query 5: Why is there a different number of genes in Table 3, for the two breeds studied?
Response: The table 3 (table 4 in the revised manuscript) depicts the DEGs that were linked with some of stress associated pathways. The intention of this table was to highlight some of these pathway associated DEGs that were expressed in the breeds due to stress and the differences in their relative expression. The differences in the numbers are because of the absence of the gene from a particular family (for example HSPs or EIFs) from the respective treatment group. The editors had suggested to re-arrange the genes presented in this table. I hope this would make things more clear now
Query 6: The following abbreviations - cs cpg chg chh - I recommend writing in a note to Table5
Response: Thank you for your suggestion, we have mentioned the details for the abbreviations in the footnote for table 6 in the revised manuscript
Query 7: In addition, I want to advise to consider splitting the article into 2 parts, because there are so many results and the information will be perceived better in separate parts.
Response: Thank you for your suggestion. We agree that the manuscript is voluminous and there is a lot of novel information in it. Our objective of putting this forth as a single manuscript was to give the readers an overall picture on how heat stress triggers responses in the caprine skin and also to highlight breed differences. However we appreciate the reviewer’s suggestions. We authors do not have any objection in splitting the manuscript into 2 provided there would not be any additional article processing charges. We do not have additional funds to support the APC for the manuscript and hence if the editors could waive it, we can work on splitting the manuscript accordingly.